# Vanillic Acid Suppresses HIF-1α Expression via Inhibition of mTOR/p70S6K/4E-BP1 and Raf/MEK/ERK Pathways in Human Colon Cancer HCT116 Cells

**DOI:** 10.3390/ijms20030465

**Published:** 2019-01-22

**Authors:** Jingli Gong, Shengxue Zhou, Shihai Yang

**Affiliations:** 1College of Chinese Medicine, Jilin Agricultural University, Changchun City 132000, China; jingligong@126.com; 2College of Chinese Medicine, Jilin Agricultural Science and Technology College, Jilin City 132101, China; ShenXue_Zhou@126.com

**Keywords:** vanillic acid, HIF-1α, angiogenesis, proliferation, antitumor activity

## Abstract

Hypoxia-inducible factor 1 (HIF-1) plays a pivotal role in tumor adaptation to microenvironmental hypoxia, and it also exerts important roles in angiogenesis and tumor development. Vanillic acid is a dietary phenolic compound reported to exhibit anticancer properties. However, the mechanisms by which vanillic acid inhibits tumor growth are not fully understood. Here, we investigated the effect of vanillic acid on HIF-1α activation. Vanillic acid significantly inhibits HIF-1α expression induced by hypoxia in various human cancer cell lines. Further analysis revealed that vanillic acid inhibited HIF-1α protein synthesis. Neither the HIF-1α protein degradation rate nor the steady-state HIF-1α mRNA levels were affected by vanillic acid. Moreover, vanillic acid inhibited HIF-1α expression by suppressing mammalian target of rapamycin/p70 ribosomal protein S6 kinase/eukaryotic initiation factor 4E-binding protein-1 and Raf/extracellular signal-regulated kinase (ERK) kinase (MEK)/ERK pathways. We found that vanillic acid dose-dependently inhibited VEGF and EPO protein expressions and disrupted tube formation. The results suggest that vanillic acid effectively inhibits angiogenesis. Flow cytometry analysis demonstrated that vanillic acid significantly induced G1 phase arrest and inhibited the proliferation of human colon cancer HCT116 cells. In vivo experiments confirmed that vanillic acid treatment caused significant inhibition of tumor growth in a xenografted tumor model. These studies reveal that vanillic acid is an effective inhibitor of HIF-1α and provides new perspectives into the mechanism of its antitumor activity.

## 1. Introduction

Hypoxia is a common phenomenon in the majority of human tumors and known to be involved in metastasis [1,2]. Under hypoxic conditions, cancer cells secrete substances to regulate their microenvironment. During tumorigenesis, hypoxia contributes to tumor cell proliferation and angiogenesis. Tumor cells promote angiogenic activity through increases in hypoxia-inducible factor 1α (HIF-1α), vascular endothelial growth factor (VEGF), erythropoietin (EPO), cyclin D1 protein expression, which are essential for tumor growth.

Hypoxia-inducible factor 1α (HIF-1α) is a heterodimeric transcription factor composed of two subunits, O2-regulated HIF-1α and constitutively expressed HIF-1β subunits [3]. In the ODDD region, HIF-1α undergoes post-transcriptional modifications under normoxia. Prolyl hydroxylases (PHDs) are crucial oxygen-sensing molecular switches. The HIF-1α protein is targeted for degradation by a PHD that uses ascorbic acid, iron, and oxygen as cofactors to hydroxylate proline residues (pro-564 and pro-402) in the presence of 2-alphaketoglutarate. Hydroxylated HIF-1α binds to the von Hippel Lindau (VHL) protein and assembles into the E3 ubiquitin ligase consisting of VHL, Cullin-2, Elongin B/C, and ring box protein 1, the VCBCR complex, and then polyubiquitination of HIF-1α targets it to the 26S proteasome for degradation [4]. Asparagine 803 in the transactivated structure domain inhibits hydroxylation of HIF-1 in fully oxygenated cells and blocks the binding of coactivators p300 and CBP. Failure to interact with coactivators can impair the ability of HIF-1 to transactivate target genes [5]. In hypoxic environments, HIF-1α accumulates because the rates of hydroxylation and ubiquitination decline. HIF-1α then heterodimerizes with HIF-1β and translocates to the nucleus to initiate its transcriptional program [6]. Stabilization of HIF-1α is mainly depends on proline hydroxylation and VHL-mediated degradation. In addition, some non-hypoxia-driven stimuli modulate HIFs, such as growth factors, cytokines, hormones, and various stressors. Hypoxia-inducible factor 1α (HIF-1α) expression is induced by several growth factors and their cognate receptors that signal through PI3K or Ras/MAPK pathways [6,7,8]. 

Vanillic acid, an oxidized form of vanillin, is a major active compound isolated from *Angelica sinensis* and green tea. Vanillin acid is a dietary phenol that can protect biofilms and inhibit lipid peroxidation in cells [9]. Vanillin acid eliminates ROS including hydroxyl radicals and lipid peroxide radicals [10]. It also has anti-microbial, anti-inflammatory, anti-cancer, and liver-protective effects [9,10,11,12,13]. In the present study, we found that vanillic acid inhibited hypoxia-induced accumulation of HIF-1α protein. Further analysis showed that reduction of HIF-1α was correlated with suppression of HIF-1α protein synthesis but not its degradation or reduction of its mRNA. The inhibitory effects of vanillic acid on HIF-1α activation were associated with suppression of rapamycin (mTOR)/p70 ribosomal protein S6 kinase (p70S6K)/eukaryotic initiation factor 4E-binding protein-1 (4E-BP1) and Raf/extracellular signal-regulated kinase (ERK) kinase (MEK)/ERK signaling pathways. On the basis of our findings, we demonstrated that vanillic acid inhibited cell proliferation through G1 phase arrest and suppressed angiogenesis. We confirmed our observations in vivo by revealing profound antitumor activity of vanillic acid in a murine xenograft model with no apparent toxicity to the animals. These data clarify the antitumor function of vanillic acid in cancer and facilitate exploring the underlying mechanisms of vanillic acid in regulating cancer development.

## 2. Results

### 2.1. Vanillic Acid Inhibits HIF-1 Transcriptional Activation

To investigate whether vanillic acid inhibited HIF-1α transcriptional activation, HCT116 cells were transfected with an HRE-dependent luciferase reporter gene and then incubated with vanillic acid. The results show that vanillic acid obviously inhibited luciferase reporter activity induced by 1% O_2_ (Figure 1B). Considering that the inhibitory effect on HIF-1α transcriptional activation may be related to vanillic acid-induced cytotoxicity, we examined cell viability. After HCT116 cells were treated with vanillic acid (up to 30 µM) for 24 h, no significant changes in cell viability were observed compared with the untreated control group (Figure 1C).

### 2.2. Vanillic Acid Inhibits HIF-1α Protein Expression Dose-Dependently 

Next, we investigated whether vanillic acid affected HIF-1α protein levels. Western blotting showed no HIF-1α protein under normoxic conditions, but it was stabilized in the 1% O_2_ or CoCl_2_ conditions and became easily detectable using Western blotting. Following 12 h of treatment, vanillic acid significantly reduced HIF-1α protein expression induced by 1% O_2_ or CoCl_2_ in HCT116 cells or SW620 cells (Figure 2A–C,F). Next, in order to confirm whether inhibition of HIF-1α by vanillic acid was specific to the cell line, we extended these experiments to different tumor cell lines, including Hep3B hepatic cancer cells and A549 human lung carcinoma cells. Figure 2D–F showed that, HIF-1α expression was significantly suppressed by vanillic acid in both cell lines under hypoxia. Vanillic acid had little effect on the protein levels of HIF-1*β* and Topo-I compared with the decrease in HIF-1α levels. We next examined the effect of vanillic acid on HIF-1α expression in HCT116 cells by immunofluorescence assays. Following 12 h of treatment, vanillic acid (30 µM) almost completely diminished nuclear protein levels of HIF-1α enhanced by hypoxia in HCT116 cells (Figure 2G).

### 2.3. Vanillic Acid Inhibits the Protein Synthesis of HIF-1α but not its Degradation

To dissect the mechanism by which vanillic acid diminishes HIF-1α protein expression, we examined whether the downregulation occurs at the transcriptional level of HIF-1α expression. Figure 3A,B show that vanillic acid did not decrease the HIF-1α mRNA level. These results imply that the vanillic acid inhibits HIF-1α protein expression might not occur at the transcriptional level. 

Generally, the accumulation of HIF-1α is dependent on the balance between its protein synthesis and degradation [14]. To determine the possible mechanism by which vanillic acid regulates HIF-1α expression, we first examined the effect of vanillic acid on HIF-1α stabilization using CHX to prevent protein synthesis. The HIF-1α protein was accumulated by exposing the cells to 1% O_2_ for 4 h and then added CHX alone or in combination with vanillic acid. Therefore, HIF-1α protein expression levels fundamentally reflected the rate of HIF-1α degradation. As shown in Figure 3C,D, HIF-1α protein levels declined quickly in the presence of CHX, and vanillic acid did not modify the HIF-1α degradation rate. These results suggest that vanillic acid might not alter HIF-1α protein stability in HCT116 cells. To further evaluate de novo protein synthesis of HIF-1α in vanillic acid-treated cells, we treated HCT116 cells with MG-132, an inhibitor of the 26S proteasome, which protected ubiquitinated HIF-1α protein from degradation. As shown in Figure 3E,F, MG-132 treatment resulted in much more accumulation of HIF-1α (lane 2 compared with lane 4). In contrast, vanillic acid inhibited HIF-1α expression by MG-132 co-treatment (see lanes 3 and 5 in Figure 3E). No effects were observed on Topo-I expression. Taken together, these results suggest that vanillic acid decreases de novo synthesis of HIF-1α protein, but does not affect its protein stability in HCT116 cells. 

### 2.4. Vanillic Acid Decreases HIF-1α Protein Synthesis Via Mtor/P70s6k/4E-BP1 and Raf/MEK/ERK Pathways in Human Colon Cancer HCT116 Cells 

Previous studies have shown that the PI3K/Akt/mTOR pathway is involved in HIF-1α protein synthesis at the translational level [15]. To confirm the potential mechanism in vanillic acid-mediated inhibition of HIF-1α expression, we examined the phosphorylation of mTOR and its effectors, p70S6K, 4E-BP1, and eIF4E. As shown in Figure 4A,E, in parallel with the alteration of HIF-1α protein expression, vanillic acid inhibited the protein levels of p-mTOR, p-p70S6K, p-4E-BP1, and p-eIF4E in a dose-dependent manner under hypoxia. Vanillic acid had almost no effect on the total protein levels for mTOR p70S6K, 4E-BP1, and eIF4E. The Raf/MEK/ERK pathway is also known to increase HIF-1α mRNA translation [16]. To confirm the inhibitory effect of vanillic acid on the Raf/MEK/ERK pathway, we examined the effect of vanillic acid on the phosphorylation of c-Raf, MEK1/2, and ERK1/2 in HCT116 cells by Western blot analysis. The results demonstrated that vanillic acid dose-dependently inhibited the expression of p-c-Raf, p-MEK1/2, and p-ERK1/2 induced by hypoxia. However, it had no inhibitory effect on total protein levels of c-Raf, MEK1/2, and ERK1/2 (Figure 4B,F). We further examined the effect of rapamycin (mTOR inhibitor), PD98059 (MEK inhibitor) on HIF-1α expression in HCT116 cells. The results showed that the vanillic acid or rapamycin suppressed the expression of HIF-1α, which was further inhibited by the combination of vanillic acid and rapamycin (Figure 4C,G). Comparable results were obtained in Figure 4F, vanillic acid or PD98059 suppressed the expression of HIF-1α, which was further inhibited by the combination of vanillic acid and PD98059 (Figure 4D,H). Taken together, the results suggest that vanillic acid inhibits HIF-1α protein expression by reducing mTOR/p70S6K/4E-BP1 and Raf/MEK/ERK pathways. 

### 2.5. Vanillic Acid Inhibits Tumor Angiogenesis

Hypoxia-inducible factor 1α (HIF-1α) mediates expressions of *VEGF* and *EPO*, important factors that involve tumor angiogenesis [17]. To determine the effect of vanillic acid on VEGF and EPO expression at protein and mRNA levels, we performed Western blotting and RT-PCR assays, respectively. Treating HCT116 cells with vanillic acid for 12 h resulted in downregulation of VEGF and EPO protein expression in a dose-dependent manner (Figure 5A,B). The expression of VEGF and EPO mRNAs in HCT116 cells was also inhibited in a dose-dependent manner (Figure 5C,D). These results indicated that vanillic acid suppressed VEGF and EPO gene and protein expression. To confirm the anti-angiogenic ability of vanillic acid, we performed a tube formation assay with human endothelial cells in vitro. The results showed that addition of vanillic acid disrupted tube formation in a dose-dependent manner (Figure 5E). These data suggest that vanillic acid has a negative effect on tumor angiogenesis.

### 2.6. Vanillic Acid Inhibits the Proliferation of Human Colon Cancer HCT116 Cells Via Blocking Cell Cycle Progression at G1 Phase

Hypoxia-inducible factor 1α (HIF-1α) mediates cell proliferation and survival [18]. Cell proliferation is primarily regulated by the cell cycle. To clarify the effect of vanillic acid on cell cycle progression in HCT116 cells, we performed flow cytometric assay to examine the DNA content of nuclei. As shown in Figure 6A, vanillic acid treatment with 30 µM markedly induced G1 phase arrest. Under hypoxia, cells in G1 phase increased from 56.96% in medium alone to 62.81, 74.62, and 83.46% in the presence of 3, 10, and 30 µM vanillic acid, respectively. These results suggested that vanillic acid suppressed cell proliferation and blocked cell cycle progression at G1 phase. Then, we elucidated the effect of vanillic acid on expression of G1 phase-regulatory proteins cyclin D1 and c-Myc. As shown in Figure 6B,C, vanillic acid dose-dependently reduced protein expression levels of cyclin D1 and c-Myc. Cell proliferation was also detected by a clonogenic assay. Following 12 h of treatment, vanillic acid (30 µM) decreased the number of hypoxia-exposed HCT116 cells and SW620 cells. In other words, the clonogenicity of HCT116 cells and SW620 cells were inhibited in a dose-dependent manner (Figure 6D,E). Furthermore, Figure 6F shows that vanillic acid or rapamycin suppressed clonogenicity of HCT116 cells, which was further inhibited by the combination of vanillic acid and rapamycin. Comparable results were obtained in Figure 6G, vanillic acid or PD98059 suppressed clonogenicity of HCT116 cells, which was further inhibited by the combination of vanillic acid and PD98059. Taken together, these results suggest that vanillic acid inhibits cell proliferation and prevents cell cycle progression at G1 phase. 

### 2.7. Vanillic Acid Inhibits the Growth of Human Colon Cancer HCT116 Cells in a Xenograft Tumor Model 

Based on our in vitro findings, HCT116 cells were injected subcutaneously in mice and induced tumor formation, administered p.o. three times a week with vanillic acid. As expected, we found that vanillic acid significantly inhibited tumor growth in vivo (Figure 7A). However, the body weight of the mice did not significantly decrease, indicating that the application of vanillic acid may display few cellular toxicity effects in vivo (Figure 7B). Tumors were harvested at 4 h after the last vanillic acid treatment, and representative tumor masses are shown in Figure 7C. Consistent with the findings in cultured cells, vanillic acid significantly reduced the HIF-1α and VEGF protein expressions in tumors tissue (Figure 7D,E). Furthermore, immunohistochemical analysis showed that vanillic acid inhibited HIF-1α and VEGF expressions compared with the control (Figure 7F). 

### 2.8. Diagram of the Proposed Mechanism by which Vanillic Acid Inhibits Cell Proliferation and Angiogenesis

In summary, we found that vanillic acid inhibited HIF-1α by suppression of mTOR/p70S6K/4E-BP1 and Raf/MEK/ERK signaling pathways, subsequently inhibiting proliferation and angiogenesis of human cancer cells (Figure 8). 

## 3. Discussion

Hypoxia-inducible factor 1α (HIF-1α) is a key stress-responsive transcription factor to low oxygen; the expression of HIF-1α is associated with tumor progression and angiogenesis. Hypoxia-inducible factor 1 (HIF-1)-induced tumor progression is correlated with oncogenic activation or loss of tumor suppressor function, such as Ha-ras, myc, src, p53, PTEN, or VHL [19]. Hypoxia-inducible factor 1α (HIF-1α) is connected with mechanisms of tumor invasion capacity, radiotherapy, and chemotherapy resistance [20]. Targeting the HIF-1α pathway is a highly promising strategy with potential for cooperation with other therapies. The discovery of small molecule HIF-1α inhibitors is an attractive approach in the development of cancer therapeutic drugs [21]. In an effort to discover small molecule inhibitors of HIF-1α, vanillic acid, a dietary phenolic compound, was identified as a potent HIF-1α inhibitor. The expression of HIF-1α can be regulated at different levels, such as HIF-1α mRNA, protein synthesis, stability, transcriptional activity, and downstream pathways [22,23]. In the present study, we demonstrated that vanillic acid significantly inhibited HIF-1α protein synthesis and transcriptional activity in hypoxic-induced human colon cancer HCT116 cells. 

The regulation of HIF-1α translation and synthesis is involved in the key upstream pathway, including the PI3K/AKT/mTOR and Raf/MEK/ERK pathways [15,16]. Both signaling pathways are related to the activation of eIF4E. Importantly, eIF4E is a pivotal translation factor; it regulates the initiation of mRNA translation [24]. The PI3K/Akt/mTOR signaling pathways have been identified as contributing to tumor progression, and which are important regulators of several functions [25]. 

The mammalian target of rapamycin (mTOR) is a serine/threonine kinase, and it carries out its functions by two distinct complexes, mTORC1 and mTORC2. For the phosphorylation of mTORC1, there are at least two downstream targets that control translation, including the 4E-BP1 and p70S6K [26]. When hypophosphorylated, 4E-BPs cooperate with eIF4E, eIF4E is prevented from binding to eIF4G, consequently blocking the formation of the eIF4F translation initiation complex. The hyper phosphorylation of 4E-BP1 by the mTORC1 complex causes its dissociation from eIF4E, which leads to enhanced initiation of translation [27]. In additional, mTORC1 phosphorylates p70S6K, leading to the phosphorylation of the ribosomal protein S6, which induces mRNA translation [28]. In the current study, we incubated human colon cancer HCT116 cells with vanillic acid, the results showed that vanillic acid inhibited p-mTOR, p-p70S6K, p-4E-BP1, and p-eIF4E protein expression levels.

Next, we explored the effect of vanillic acid on the Raf/MEK/ERK pathway that directly phosphorylates eIF4E and plays a pivotal role in tumor cell survival and proliferation [29]. As shown in Figure 4C, vanillic acid significantly suppressed the phosphorylation of Raf, MEK, and ERK in human colon cancer HCT116 cells. Additionally, Western blot assay showed that rapamycin or PD98059 suppressed the expression of HIF-1α, which was further inhibited by co-treatment with vanillic acid. Combined, these data indicate that mTOR/p70S6K/4E-BP1 and Raf/MEK/ERK pathways play a key role in vanillic acid-mediated inhibition of HIF-1α.

The transcription of VEGF and EPO genes are mediated by HIF-1α; they are hypoxia-inducible neuroprotective cytokines [30]. In tumor tissues, *VEGF* is essential for tumor angiogenesis; it can induce the new blood vessels formation [31]. Under hypoxia conditions, HIF-1α bound to the hypoxic response element in its 5′ flanking region for the transcriptional activation, leading to the VEGF protein expression increases [32]. *EPO*, a pleiotropic cytokine mainly secreted by the kidneys, stimulates the production of red blood cells [33]. Erythropoietin (EPO) plays an important role in tumor angiogenesis in vivo; it can promote tumor cells survival and growth under hypoxic [34]. Since EPO was induced by HIF-1α, we further examined whether vanillic acid suppressed EPO and VEGF transcription in human colon cancer HCT116 cells. As expected, vanillic acid decreased VEGF and EPO protein and mRNA levels. Moreover, the tube formation assay showed that vanillic acid mediated antitumor effects through suppression of angiogenesis.

Vanillic acid was found to decrease cell proliferation by cell G1 phase arrest. Cyclin D1 is a key protein in the regulation of the cell cycle at the G1 to S phase transition; it is essential for regulation of proliferation, differentiation, and transcriptional control [35]. Cellular-myelocytomatosis viral oncogene (c-Myc) is a basic-helix-loop-helix/leucine zipper transcription factor. In the majority of human tumors, c-Myc is abnormal expression that controls the cell cycle transition. [36]. Therefore, unsurprisingly, we observed that vanillic acid downregulated the protein levels of cyclin D1 and c-Myc. Comparable results were also obtained in the clonogenic assay with vanillic acid significantly inhibiting cell growth. Additionally, clonogenic assay also showed that rapamycin or PD98059 suppressed clonogenicity of HCT116 cells, which was further inhibited by co-treatment with vanillic acid.

## 4. Materials and Methods 

### 4.1. Cell Culture and Reagents

The HCT116 cells were grown in RPMI with penicillin (100 units/mL)-streptomycin (100 units/mL) (Invitrogen, Carlsbad, CA, USA) and 10% heat-inactivated fetal bovine serum (Hyclone, Logan, UT, USA). The Hep3B, A549, and HUVEC cells were maintained in DMEM medium supplemented as above. All cells were purchased from American Type Culture Collection (ATCC, Manassas, VA, USA). The hypoxic culture was kept in a gas-controlled chamber (Thermo Electron Corp., Marietta, OH, USA) maintained at 1% O_2_, 94% N_2_, and 5% CO_2_ at 37 °C. The MG-132 and cycloheximide (CHX) were bought from Sigma Chemical Co (St Louis, MO, USA). The mTOR inhibitor (rapamycin) and MEK inhibitor (PD98059) were obtained from Calbiochem (San Diego, CA, USA). Vanillic acid was purchased from Yuanye Biotechnology Co. Ltd. (Shanghai, China) and its structure is shown in Figure 1A. The purity of vanillic acid was more than 98% in HPLC analysis. 

### 4.2. Luciferase Reporter Assay

The transcriptional activity of the HIF-1α was confirmed using a luciferase assay. The pGL3-HRE-Luciferase plasmid containing six copies of HREs derived from the human *VEGF* gene and with pRL-CMV was transfected into the cells (Promega, Madison, WI, USA). Following 24 h of transfection, cells were incubated with vanillic acid for an additional 12 h under normoxic and hypoxic conditions. The luciferase activity was measured by Microlumat plus luminometer (EG&G Berthold, Bad Wildbad, Germany) via injecting 100 µL of luciferin-containing assay buffer and measuring luminescence for 10 s. The experiments were performed in triplicate, and similar results were obtained from at least three independent experiments.

### 4.3. 3-(4,5-Dimethylthiazol-2-yl)-2,5-diphenyltetrazolium bromide (MTT) Assay

The relative cell viabilities were determined by the MTT assay (Sigma-Aldrich, St. Louis, MO, USA). Briefly, cells were seeded onto a 96-well culture plate at a density of 1 × 10^4^ cells/well in 100 µL of complete RPMI. Then cells were treated with different concentrations of vanillic acid for 12 h. Next, MTT was added to each well and incubated for 4 h. Media were then discarded and 100 µL of DMSO (Sigma) was added. The absorbance was detected by Multiskan GO at 570 nm (Thermo Electron Corp., Marietta, OH, USA).

### 4.4. Western Blot Analysis

Whole cell extracts were prepared by lysing cells in an ice-cold lysis buffer (50 mM Tris-HCl, pH 7.5, 1% Nonidet P-40, 1 mM EDTA, 1 mM phenylmethyl sulfonylfluoride) containing protease inhibitor cocktail. According to the manufacturer’s instructions, the nuclear extracts were obtained with NE-PER reagent (Pierce, Rockford, IL, USA). The protein concentrations were determined using the Bradford method. Equal amounts of protein were subjected to SDS-PAGE and electrophoretically transferred onto a PVDF membrane. Membranes were blocked with 5% skimmed milk in PBST for 1 h at room temperature with shaking and incubated with indicated primary antibodies. Antibody for HIF-1α (610959) was obtained from BD Biosciences (San Diego, CA, USA). Antibodies for phospho-mTOR (2971), mTOR (2972), phospho-p70S6K (9204), phospho-4E-BP1 (9451), phospho-eIF4E (9741), phospho-c-Raf (9431), phospho-MEK1/2 (9154), phospho-ERK1/2 (4370) and cyclin D1 (2922) were purchased from Cell Signaling Technology (Beverly, MA, USA). Antibodies for p70S6K (sc-8418), 4E-BP1 (sc-6936), eIF4E (sc-9976), c-Raf (sc-24560), MEKK1/2 (sc-449), ERK1/2 (sc-514302), HIF-1*β* (sc-17811), topoisomerase-I (Topo-I, sc-5342), VEGF (sc-1542) were obtained from Santa Cruz Biotechnology (Santa Cruz, CA, USA). Antibody for EPO (ab129452) was from Abcam (Cambridge, MA, USA). Antibody for α-tubulin (T5168) was from Sigma–Aldrich. Antibody for c-Myc (P50173M) was from Abmart (Shanghai, China). The membranes were then incubated with a horseradish peroxidase-conjugated secondary antibody. Proteins were detected by enhanced chemiluminescence.

### 4.5. Immunofluorescence Assay

The cells were plated on coverslips, the cells were washed with PBS, and then fixed with 4% paraformaldehyde (Sigma–Aldrich) for 15 min and permeabilized with 0.2% Triton X-100, and then blocked with 5% BSA (Sigma-Aldrich) in PBS for 30 min. Fixed cells were then blocked with 5% BSA in PBS to reduce non-specific immune reactivity, and the cells were incubated with HIF-1α antibodies overnight at 4 °C. After washing 3 times with PBS, the cells were incubated with Alexa flour^®^ 488 goat anti-mouse lgG (H+L) for 30 min at room temperature, then with DAPI (4′,6-diamidino-2-phenylindole) for 30 min before observation. The fluorescently stained cells were then examined using a microscope (Olympus, Japan).

### 4.6. RT-PCR Analysis

Total RNA was extracted from the HCT116 cells using the Qiagen RNeasy Mini kit (Qiagen, Valencia, CA, USA). Total RNA (2 µg) was used to perform RT-PCR using RT-PCR kit (Invitrogen, Carlsbad, California, USA). The sequences of the primers used were as follows: for HIF-1α-F, 5′-CTCAAAGTCCGACAGCCTCA-3′; for HIF-1α-R, 5′-CCCTGCAGTAGGTTTCTGCT-3′; for GAPDH-F, 5′-ACCACAGT CCATGCCATCAC-3′; for GAPDH-R, 5′-TCCACCACCCTGTTGCTGTA-3′; for VEGF-F, 5′-GCTCTACCTCCACCATGCCAA-3′; for VEGF-R, 5′-TGGAAGATGTC CACCAGGGTC-3′; for EPO-F, 5′-CACTTTCCGCAAACTCTTCCG-3′; for EPO-R, 5′-GTCACAGCTTGCCACCTAAG-3′. Following PCR, reaction products were resolved on 2% agarose gels, stained with ethidium bromide, and photographed under UV light.

### 4.7. Tube Formation Assays

Chilled liquid matrigel was dispensed onto 24-well plates (300 µL per well), and allowed to solidify for 1 h at 37 °C according to the manufacturer’s instructions. Human Umbilical Vein Endothelial cells (HUVEC) (2.5 × 10^4^ cells/well) suspended in 300 µL of fresh DMEM medium were added to each well coated with matrigel and treated with or without vanillic acid for 12 h. Tube formation and capillary tube lengths were observed under a microscope and photographed (Olympus, Tokyo, Japan).

### 4.8. Clonogenic Assay

The 3 × 10^3^ cells/well of HCT116 cells were seeded in 6-well plates for 24 h before exposure to drug treatment. After treatment with vanillic acid for 12 h, the medium was discarded and the cells were washed with 1 × PBS and exchanged with medium containing 10% FBS. After ten days, the cells were washed with PBS. The colonies were stained with 1% crystal violet for 30 s after fixation with 10% formaldehyde for 5 min. Representative images were photographed with a digital camera (Nikon, Japan).

### 4.9. Flow Cytometry Analysis

Unsynchronized cells were exposed to vanillic acid for 12 h and harvested from culture dishes. After washing with PBS, the cells suspended in 70% ethanol solution and kept at −20 °C overnight. Prior to analysis, cells were washed twice in PBS, resuspended in staining solution (final concentration 0.1% Triton X-100, 0.5 mg/mL RNase A and 0.025 mg/mL PI), incubated in the dark at room temperature for 30 min. The DNA content was analyzed using a FACSCalibur flow cytometer with Cell Quest software (Becton-Dickinson, Franklin Lakes, NJ, USA). The ModFit LT V4.0 software package (Verity Software, Topsham, ME, USA) was used to analyze the data.

### 4.10. Xenografted Tumor Model

All experimental procedures were approved by the Institutional Animal Care and Use Committee of Jilin Agricultural University. The male Balb/c nude mice (4–5 weeks of age, 20 ± 2 g, Vital River Laboratory Animal, Beijing, China) were randomly divided into 3 groups (*n* = 5/group). For tumor cell implantation, HCT116 cells were harvested and 1 × 10^7^ cells in 200 µL phosphate-buffered saline were subcutaneously injected into the mice. After five days, mice were treated with or without vanillic acid (10 and 30 mg/kg) three times a week. Tumor size was measured in two dimensions with calipers every 5 days, up to 50 days after injection. Tumor volume was calculated using the equation: (length × (width)^2^)/2. The mice were killed by cervical dislocation and solid tumors were removed. The tumor tissues were harvested and used for Western blot and immunohistochemistry.

### 4.11. Immunohistochemical Analysis

Tumor tissues were fixed in 10% neutral-buffered formalin for 24 h, processed, and embedded in paraffin blocks. Hematoxylin and eosin (H&E) staining, and immunohistostaining were performed on 4–5 µm-thick paraffin-embedded sections. Sections were deparaffinized and incubated in 3% H_2_O_2_ for 10 min to quench endogenous peroxidase activity. After blocking with normal goat serum for 30 min, the sections were incubated with anti-HIF-1α monoclonal antibody and anti-VEGF polyclonal antibody overnight at 4 °C. Subsequently, the sections were incubated with biotin-conjugated secondary antibody for 2 h at room temperature, and horseradish peroxidase streptavidin for 1 h at room temperature. The level of histopathological changes and positive stained area in tumor tissues was observed by a light microscope.

### 4.12. Statistical Analysis

Results are expressed as mean ± SD. A comparison of the results was performed with one-way ANOVA and Tukey’s multiple comparison tests (Graphpad Software, Inc, San Diego, CA, USA). Differences were considered significant when *p*-values were smaller than 0.05.

## 5. Conclusions

In conclusion, our present study demonstrated that vanillic acid decreases HIF-1α protein synthesis by inhibiting mTOR/p70S6K/4E-BP1 and Raf/MEK/ERK pathways in human colon cancer HCT116 cells. Moreover, vanillic acid inhibited angiogenesis and cell proliferation which are essential for cancer cells to adapt to the microenvironmental hypoxia and tumor progression. The present study also suggests the potential use of vanillic acid cancer treatment, and thus provides a basis for the development of vanillic acid as an anticancer drug.

## Figures and Tables

**Figure 1 ijms-20-00465-f001:**
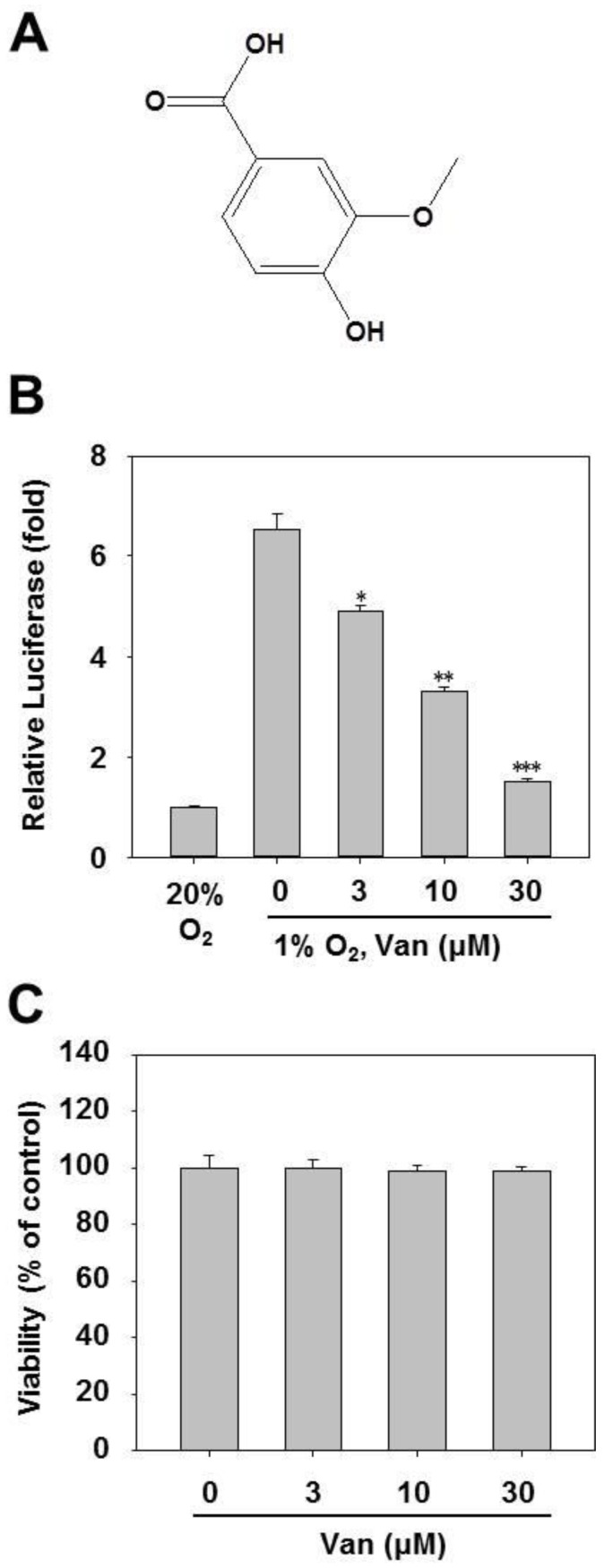
Identification of vanillic acid (Van) as a HIF-1 pathway inhibitor from a cell-based screening assay. (**A**) Chemical structure of vanillic acid (Van). (**B**) HCT116 cells were transiently co-transfected with a pGL3-HRE-Luciferase and pRL-CMV vectors. Following 24 h incubation, cells were treated with various concentrations of vanillic acid (Van) and then subjected to hypoxia, or remained in normoxia for 12 h. Data were shown as mean ± SD (*n* = 3). * *p* < 0.05, ** *p* < 0.01, *** *p* < 0.001, compared with hypoxia control. (**C**) Cells were incubated with different concentrations of vanillic acid (Van). After 24 h incubation, cell viability was determined by MTT assays.

**Figure 2 ijms-20-00465-f002:**
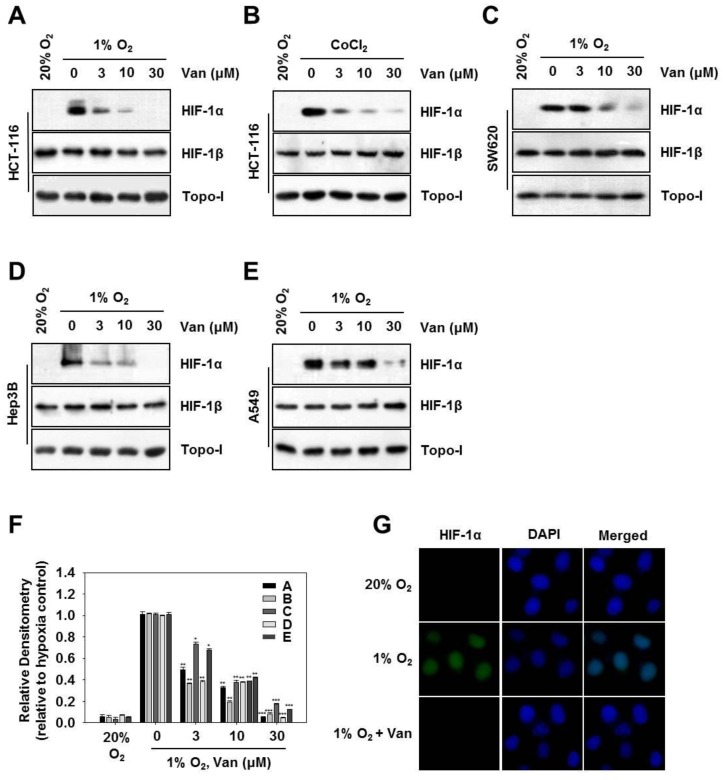
Vanillic acid (Van) inhibits HIF-1α protein expression in a dose-dependent manner. (**A**,**C**–**E**) HCT116, SW620 cells, Hep3B, and A549 cells were pretreated without or with indicated concentration of vanillic acid (Van), then cultured under normoxic or hypoxic conditions for 12 h. Whole-cell lysates for HIF-1*β* and nuclear extract for HIF-1α were detected by Western blot. Anti-Topo-I antibody was used as a loading control. (**B**) HCT116 cells were cultured with the indicated concentration of vanillic acid (Van) for 30 min and treated with CoCl_2_ (200 µM). After 12 h incubation, the whole-cell lysates for HIF-1β and nuclear extract for HIF-1α was detected by Western blot. Anti-Topo-I antibody was used as a loading control. (**F**) Data were shown as mean ± SD (*n* = 3). * *p* < 0.05, ** *p* < 0.01, *** *p* < 0.001, compared with hypoxia control. (**G**) HCT116 cells cultured in chamber slides, then were treated with or without vanillic acid (Van, 30 µM) under normoxic or hypoxic conditions for 12 h. The left column shows the HIF-1α protein in green fluorescence. The middle column shows nuclei stained with 4′,6-diamidino-2-phenylindole (DAPI, blue fluorescence). The right column shows the merged images. Magnification = 400×.

**Figure 3 ijms-20-00465-f003:**
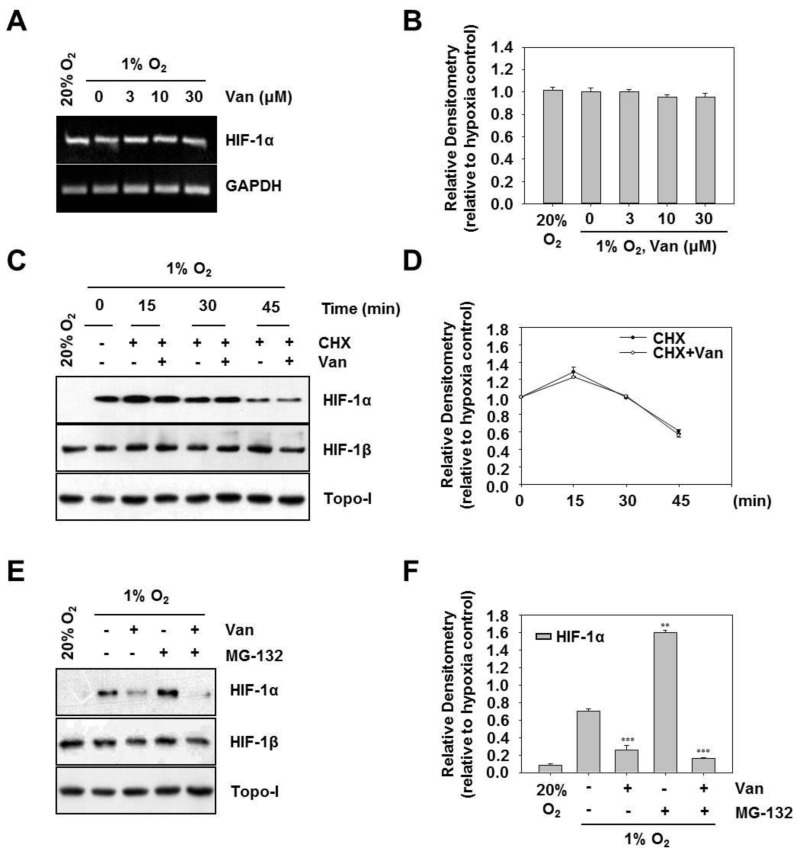
Vanillic acid (Van) inhibits the protein synthesis of HIF-1α but not its degradation. (**A**) HCT116 cells were pretreated without or with indicated concentration of vanillic acid (Van), then cultured under normoxic or hypoxic conditions for 12 h. Total RNA was analyzed by RT-PCR as described in “Materials and Methods”. (**B**) Data were shown as mean ± SD (*n* = 3). (**C**) HCT116 cells were first treated under hypoxia for 4 h. Then cells were incubated in the presence of cycloheximide (CHX, 10 µM) and vanillic acid (Van, 30 µM). After 15, 30, or 45 min following the addition of cycloheximide (CHX), the whole-cell lysates for HIF-1β and nuclear extract for HIF-1α was detected by Western blot. Anti-Topo-I antibody was used as a loading control. (**D**) Data were shown as mean ± SD (*n* = 3). (**E**) HCT116 cells were cultured with proteasome inhibitor MG-132 (10 µM) for 30 min and treated with vanillic acid (Van, 30 µM). Then cells were transferred to hypoxia condition. The whole-cell lysates for HIF-1β and nuclear extract for HIF-1α was detected by Western blot. Anti-Topo-I antibody was used as a loading control. (**F**) Data are shown as mean ± SD (*n* = 3). ** *p* < 0.01, *** *p* < 0.001, compared with hypoxia control.

**Figure 4 ijms-20-00465-f004:**
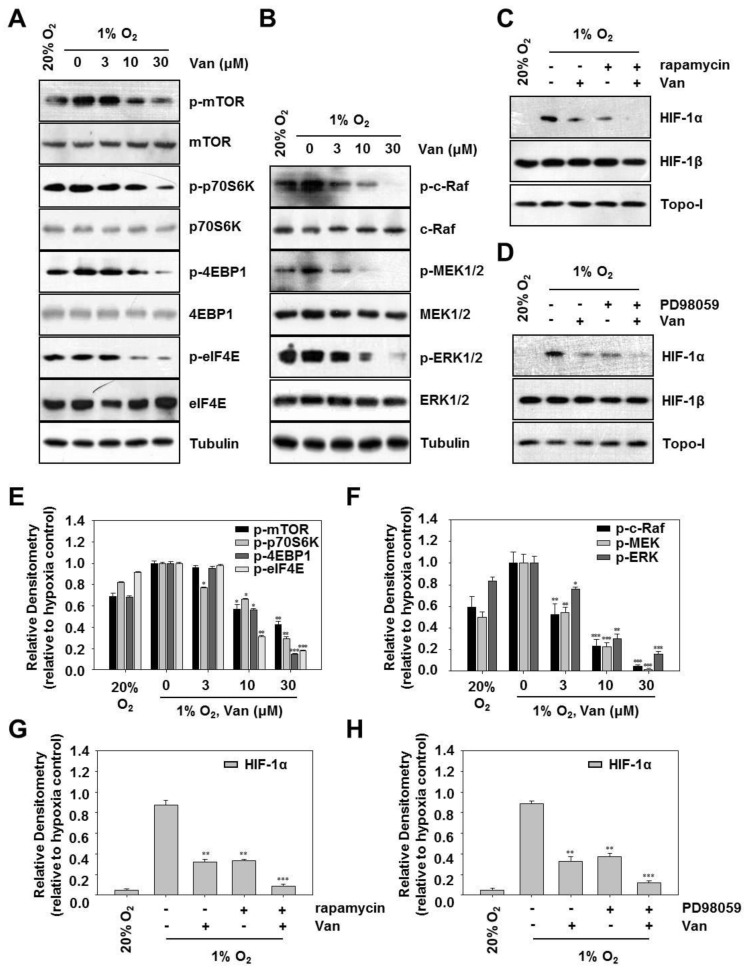
Vanillic acid (Van) decreased HIF-1α protein synthesis via mTOR/p70S6K/4E-BP1 and Raf/MEK/ERK pathways in HCT116 cells. (**A**) HCT116 cells were pretreated without or with indicated concentration of vanillic acid (Van), then cultured under normoxic or hypoxic conditions. After 12 h incubation, phospho-mTOR, phospho-p70S6K, phospho-4E-BP1, and phospho-eIF4E were detected by Western blot analysis. The bottom represents the corresponding total protein to show the equal loading of cell lysates. Levels of tubulin were used as a loading control. (**E**) Data are shown as mean ± SD (*n* = 3). * *p* < 0.05, ** *p* < 0.01, *** *p* < 0.001, compared with hypoxia control. (**B**) HCT116 cells were pretreated without or with indicated concentration of vanillic acid (Van), then cultured under normoxic or hypoxic conditions. After 12 h incubation, phospho-c-Raf, phospho-MEK1/2, and phospho-ERK1/2 were detected by Western blot analysis. The bottom represents the corresponding total protein to show the equal loading of cell lysates. Levels of tubulin were used as a loading control. (**F**) Data are shown as mean ± SD (*n* = 3). * *p* < 0.05, ** *p* < 0.01, *** *p* < 0.001, compared with hypoxia control. (**C**,**D**) HCT116 cells were pretreated without or with indicated concentration of vanillic acid (30 µM), rapamycin (100 nM), and PD98059 (50 µM), then cultured under normoxic or hypoxic conditions. After cells were harvested, the whole-cell lysates for HIF-1β and nuclear extract for HIF-1α was detected by Western blot. Anti-Topo-I antibody was used as a loading control. (**G**,**H**) Data were shown as mean ± SD (*n* = 3). ** *p* < 0.01, *** *p* < 0.001, compared with hypoxia control.

**Figure 5 ijms-20-00465-f005:**
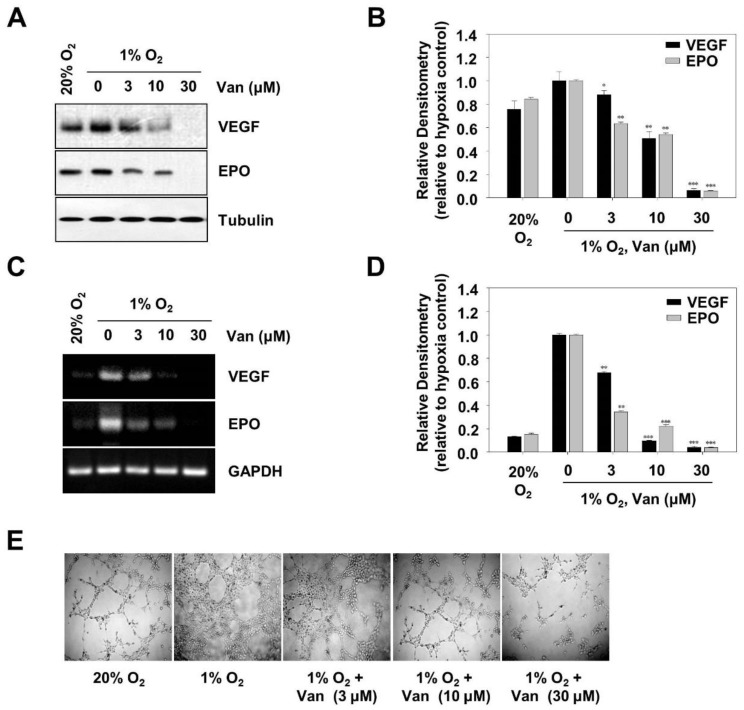
Vanillic acid (Van) inhibits tumor angiogenesis. (**A**) HCT116 cells were pretreated without or with indicated concentration of vanillic acid (Van), then cultured under normoxic or hypoxic conditions. After 12 h incubation, VEGF and EPO were detected by Western blot analysis. Levels of tubulin were used as a loading control. (**B**) Data were shown as mean ± SD (*n* = 3). * *p* < 0.05, ** *p* < 0.01, *** *p* < 0.001, compared with hypoxia control. (**C**) HCT116 cells were pretreated without or with indicated concentration of vanillic acid (Van), then cultured under normoxic or hypoxic conditions for 12 h. Total RNA was analyzed by RT-PCR as described in “Materials and Methods”. (**D**) Data are shown as mean ± SD (*n* = 3). ** *p* < 0.01, *** *p* < 0.001, compared with hypoxia control. (**E**) Human Umbilical Vein Endothelial Cells (HUVEC) were pretreated without or with indicated concentration of vanillic acid (Van), then cultured under normoxic or hypoxic conditions. The effect of vanillic acid (Van) on HUVEC cells tube formation was observed under a microscope. Magnification = 100×.

**Figure 6 ijms-20-00465-f006:**
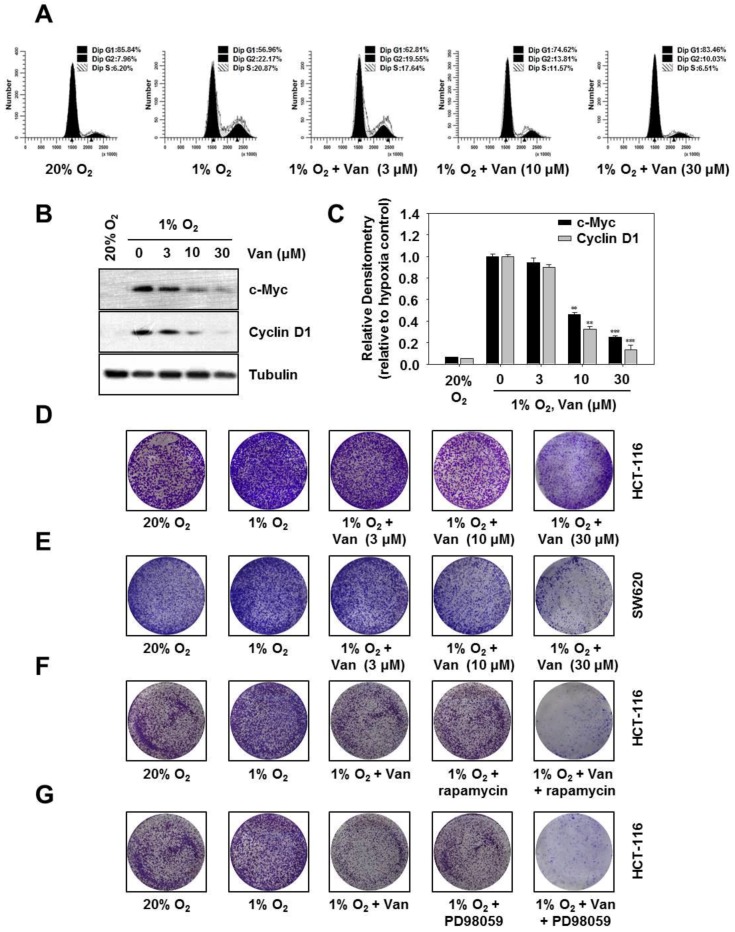
Vanillic acid inhibits the proliferation of HCT116 cells via blocking cell cycle progression in the G1 phase. (**A**) HCT116 cells were pretreated without or with indicated concentration of vanillic acid (Van), then cultured under normoxic or hypoxic conditions for 12 h. Subsequently, cell cycle status was analyzed by by FACSCalibur flow cytometry. (**B**) HCT116 cells were pretreated without or with indicated concentration of vanillic acid (Van), then cultured under normoxic or hypoxic conditions for 12 h. Cyclin D1 and c-Myc were detected by Western blot analysis. Levels of tubulin were used as a loading control. (**C**) Data were shown as mean ± SD (*n* = 3). ** *p* < 0.01, *** *p* < 0.001, compared with hypoxia control. (**D**,**E**) HCT116 cells and SW620 cells were pretreated without or with indicated concentration of vanillic acid (Van), then cultured under normoxic or hypoxic conditions for 12 h, and then the medium was replaced by fresh medium. Cells were allowed to grow for 10 d in normoxic or hypoxic conditions. (**F**,**G**) HCT116 cells were pretreated without or with indicated concentration of vanillic acid (30 µM), rapamycin (100 nM), and PD98059 (50 µM), cultured under normoxic or hypoxic conditions, and then the medium was replaced by fresh medium. Cells were allowed to grow for 10 d.

**Figure 7 ijms-20-00465-f007:**
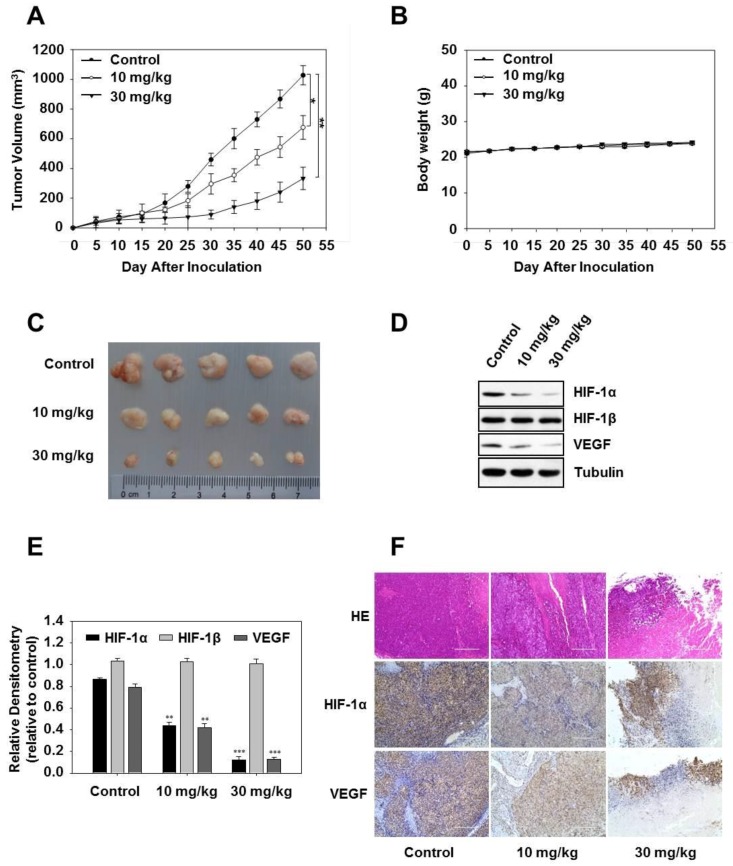
Vanillic acid inhibits tumor growth in a human tumor model. (**A**) and (**B**) HCT-116 cells were implanted subcutaneous injection (s.c) in the left flanks of nude mice, which were administered per oral (p.o) three times a week with vehicle (*n* = 5), vanillic acid (Van, 10 mg/kg, *n* = 5) or vanillic acid (Van, 30 mg/kg, *n* = 5) starting from day five. Tumor volume and mouse weight were measured using an equation and a digital balance, respectively, every five days. (**C**) Representative tumor masses of 3 groups, which were harvested 4 h after the last treatment, and the isolated tumors were photographed. (**D**) Western blot analysis of HIF-1α, HIF-1β, and VEGF in tumor blocks is shown. Tubulin was used as a loading control. (**E**) Data are shown as mean ± SD (*n* = 3). ** *p* < 0.01, *** *p* < 0.001, compared with hypoxia control. (**F**) Tumor sections were stained with hematoxylin and eosin and analyzed by immunohistochemistry for HIF-1α and VEGF. Original magnification, 200×.

**Figure 8 ijms-20-00465-f008:**
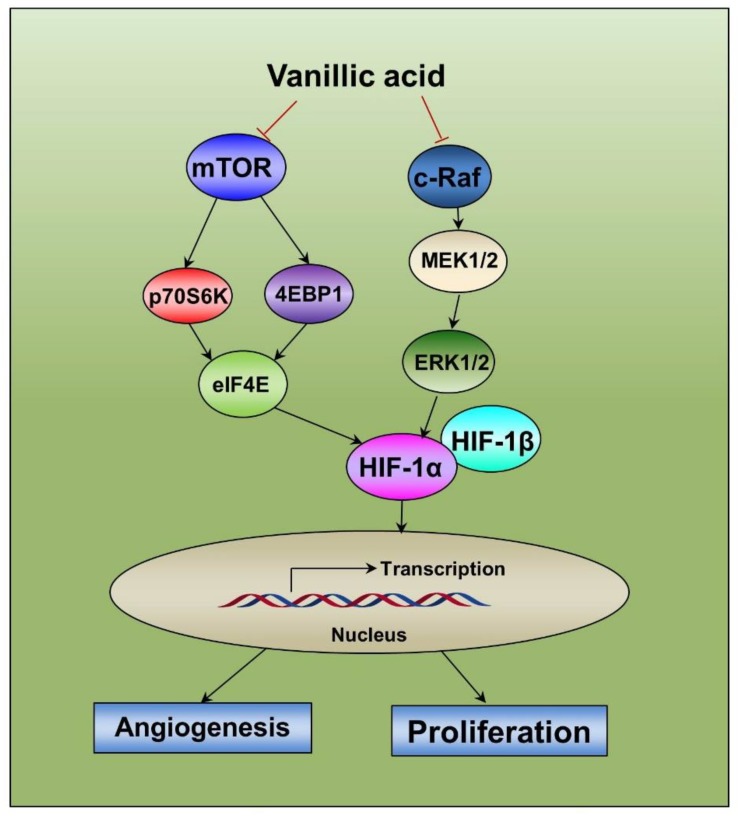
Diagram of the proposed mechanism by which vanillic acid (Van) inhibits proliferation and angiogenesis. The anti-proliferation and anti-angiogenesis effect of vanillic acid (Van) occurs mainly through the downregulation of HIF-1α by the inhibition of the mTOR/p70S6K/4E-BP1 and Raf/MEK/ERK signaling pathways. This study indicates that HIF-1α can be a potential anti-cancer target for vanillic acid (Van).

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
