# Peer review of "Vanillic Acid Suppresses HIF-1α Expression via Inhibition of mTOR/p70S6K/4E-BP1 and Raf/MEK/ERK Pathways in Human Colon Cancer HCT116 Cells"

_ijms, 2019, doi:10.3390/ijms20030465_

Round 1
Reviewer 1 Report
Authors suggest that Vanillic acid significantly inhibited HIF-1α expression via inhibition of mTOR/p70S6K/4E-BP1 and Raf/MEK/ERK pathways in human colon cancer cells. Despite many data, it has some concerns as follows:
In general, the mechanistic study of this MS is not deep despite many data. Therea are lacking evidences for title. How about effect of mTOR/p70S6K/4E-BP1 and Raf/MEK/ERK inhibitor or siRNA transfection on proliferation and HIF1 alpha in HCT116 cells under hypoxia?
Show antiproliferative effect of vanillic acid in other colon cancer lines under hypoxia. You just showed its effect only on HCT116 cells.
Add significance between lane 4 and lane 4 in Fig3 F
Method of tube formation assay is not clear. Did you add media supernatant on HUVEcs?
Rewrite Abstract including G1 arrest and VEGF data
Author Response
Thank you very much for decision letter along with the reviewer’s comments for our manuscript No.: ijms-414213
The reviewers requested modification, and more clarification. They also gave us several excellent suggestions which would strengthen our manuscript. We thank for reviewer’s constructive criticisms.
Please find some of our changes in revised MS. as highlighted in red in marked version.
Our point by point response to the comments of the reviewer is as follows:
Reviewer 1:
1. In general, the mechanistic study of this MS is not deep despite many data. There are lacking evidences for title. How about effect of mTOR/p70S6K/4E-BP1 and Raf/MEK/ERK inhibitor or siRNA transfection on proliferation and HIF1 alpha in HCT116 cells under hypoxia?
Response 1: Thanks to the reviewer’s suggestions. Previously, there are many papers that have proved that mTOR/p70S6K/4E-BP1 and Raf/MEK/ERK inhibitor or siRNA transfection can reduced the proliferation and HIF-1α expression [1-3]. The aim of our article is to study nature products which have potential anti-cancer effects and we have proved that vanillic acid could be a profound lead compound for further study. In addition, we have referenced articles that study the effect of drugs on proliferation and HIF-1α signaling pathway, while these articles have not study the effect of mTOR/p70S6K/4E-BP1 and Raf/MEK/ERK inhibitor or siRNA on proliferation and HIF-1α expression in cancer cells [4-7]. We will exam the mTOR/p70S6K/4E-BP1 and Raf/MEK/ERK inhibitor or siRNA for in-depth mechanism study in the next paper when we contribute to your journal.
The references are as below.
1. Moschetta, M. G.; Leonel, C.; Maschio-Signorini, L. B.; Borin, T. F.; Gelaleti, G. B.; Jardim-Perassi, B. V.; Ferreira, L. C.; Sonehara, N. M.; Carvalho, L. G. S.; Hellmen, E.; de Campos Zuccari, D. A. P., Evaluation of Angiogenesis Process After Metformin and LY294002 Treatment in Mammary Tumor. Anti-cancer agents in medicinal chemistry 2018.
2. Li, J.; Mi, C.; Ma, J.; Wang, K. S.; Lee, J. J.; Jin, X., Dihydrotanshinone I inhibits the translational expression of hypoxia-inducible factor-1alpha. Chemico-biological interactions 2015, 240, 48-58.
3. Shen, K.; Ji, L.; Gong, C.; Ma, Y.; Yang, L.; Fan, Y.; Hou, M.; Wang, Z., Notoginsenoside Ft1 promotes angiogenesis via HIF-1alpha mediated VEGF secretion and the regulation of PI3K/AKT and Raf/MEK/ERK signaling pathways. Biochemical pharmacology 2012, 84, (6), 784-92.
4. Mi, C.; Ma, J.; Wang, K. S.; Zuo, H. X.; Wang, Z.; Li, M. Y.; Piao, L. X.; Xu, G. H.; Li, X.; Quan, Z. S.; Jin, X., Imperatorin suppresses proliferation and angiogenesis of human colon cancer cell by targeting HIF-1alpha via the mTOR/p70S6K/4E-BP1 and MAPK pathways. Journal of ethnopharmacology 2017, 203, 27-38.
5. Ma, J.; Han, L. Z.; Liang, H.; Mi, C.; Shi, H.; Lee, J. J.; Jin, X., Celastrol inhibits the HIF-1alpha pathway by inhibition of mTOR/p70S6K/eIF4E and ERK1/2 phosphorylation in human hepatoma cells. Oncology reports 2014, 32, (1), 235-42.
6. Zhang, L.; Chen, C.; Duanmu, J.; Wu, Y.; Tao, J.; Yang, A.; Yin, X.; Xiong, B.; Gu, J.; Li, C.; Liu, Z., Cryptotanshinone inhibits the growth and invasion of colon cancer by suppressing inflammation and tumor angiogenesis through modulating MMP/TIMP system, PI3K/Akt/mTOR signaling and HIF-1alpha nuclear translocation. International immunopharmacology 2018, 65, 429-437.
7. Lim, J. H.; Lee, E. S.; You, H. J.; Lee, J. W.; Park, J. W.; Chun, Y. S., Ras-dependent induction of HIF-1alpha785 via the Raf/MEK/ERK pathway: a novel mechanism of Ras-mediated tumor promotion. Oncogene 2004, 23, (58), 9427-31.
2. Show antiproliferative effect of vanillic acid in other colon cancer lines under hypoxia. You just showed its effect only on HCT116 cells?
Response 2: We thank the reviewer’s suggestions.
Refer to the following articles, using singly HCT116 cell line might adequately prove the anti-proliferative effect of active compound [1-3], and HCT116 cells is a representative and common used cell line for the research of colon cancer.
In addition, referred to the title of the following articles, we have revised the title “vanillic acid suppresses HIF-1α expression via inhibition of mTOR/p70S6K/4E-BP1 and Raf/MEK/ERK pathways in human colon cancer cells” to “Vanillic acid suppresses HIF-1α expression via inhibition of mTOR/p70S6K/4E-BP1 and Raf/MEK/ERK pathways in human colon cancer HCT116 cells”[4-6].
For above reasons, we chose HCT116 cells for subsequent assays in our study.
The references are as below.
1. Mi, C.; Ma, J.; Wang, K. S.; Zuo, H. X.; Wang, Z.; Li, M. Y.; Piao, L. X.; Xu, G. H.; Li, X.; Quan, Z. S.; Jin, X., Imperatorin suppresses proliferation and angiogenesis of human colon cancer cell by targeting HIF-1alpha via the mTOR/p70S6K/4E-BP1 and MAPK pathways. Journal of ethnopharmacology 2017, 203, 27-38.
2. Ma, J.; Li, J.; Wang, K. S.; Mi, C.; Piao, L. X.; Xu, G. H.; Li, X.; Lee, J. J.; Jin, X., Perillyl alcohol efficiently scavenges activity of cellular ROS and inhibits the translational expression of hypoxia-inducible factor-1alpha via mTOR/4E-BP1 signaling pathways. International immunopharmacology 2016, 39, 1-9.
3. Peng, W.; Zhang, S.; Zhang, Z.; Xu, P.; Mao, D.; Huang, S.; Chen, B.; Zhang, C.; Zhang, S., Jianpi Jiedu decoction, a traditional Chinese medicine formula, inhibits tumorigenesis, metastasis, and angiogenesis through the mTOR/HIF-1alpha/VEGF pathway. Journal of ethnopharmacology 2018, 224, 140-148.
4. Wang, H. G.; Cao, B.; Zhang, L. X.; Song, N.; Li, H.; Zhao, W. Z.; Li, Y. S.; Ma, S. M.; Yin, D. J., KLF2 inhibits cell growth via regulating HIF-1alpha/Notch-1 signal pathway in human colorectal cancer HCT116 cells. Oncology reports 2017, 38, (1), 584-590.
5. Asif, M.; Shafaei, A.; Abdul Majid, A. S.; Ezzat, M. O.; Dahham, S. S.; Ahamed, M. B. K.; Oon, C. E.; Abdul Majid, A. M. S., Mesua ferrea stem bark extract induces apoptosis and inhibits metastasis in human colorectal carcinoma HCT 116 cells, through modulation of multiple cell signalling pathways. Chinese journal of natural medicines 2017, 15, (7), 505-514.
6. Ding, Z.; Xu, F.; Tang, J.; Li, G.; Jiang, P.; Tang, Z.; Wu, H., Physcion 8-O-beta-glucopyranoside prevents hypoxia-induced epithelial-mesenchymal transition in colorectal cancer HCT116 cells by modulating EMMPRIN. Neoplasma 2016, 63, (3), 351-61.
3. Add significance between lane 4 and lane 4 in Fig3 F.
Response 3: We thank the reviewer’s suggestions.The intensity of the protein bands from the western blot experiments were quantified again. As shown in Fig3 F, lane 4 have significant differences compared with lane 2. We have revised the figures as below:
4. Method of tube formation assay is not clear. Did you add media supernatant on HUVEcs?
Response 4: We thank the reviewer’s suggestions. In the tube formation assay, we added DMEM medium on HUVEC cells. In other words, HUVEC cells suspended in 300 μl of fresh DMEM medium were added to each well coated with matrigel and treated with or without vanillic acid for 12 h. We have added the method of tube formation assay in the manuscript. We have revised the method as below:
We have revised “Chilled liquid matrigel was dispensed onto 24-well plates and allowed to solidify. Then HUVEC cells were seeded onto the gel and cultured in the medium containing vanillic acid at 37°C for 12 h. Matrigel was fixed, and examined under inverted microscope (Olympus, Tokyo, Japan)” to “Chilled liquid Matrigel was dispensed onto 24-well plates (300 μl per well), and allowed to solidify for 1 h at 37 °C according to the manufacturer’s instructions. HUVEC cells (2.5 × 104 cells/well) suspended in 300 μl of fresh DMEM medium were added to each well coated with matrigel and treated with or without vanillic acid for 12 h. Tube formation and capillary tube lengths were observed under a microscope and photographed (Olympus, Tokyo, Japan)” (Page 4, Line 154 to 158).
5. Rewrite Abstract including G1 arrest and VEGF data.
Response 5: We thank the reviewer’s suggestions. We have rewrited abstract including G1 arrest and VEGF data in the manuscript. We have revised the contents as below:
We have revised “Furthermore, we found that vanillic acid inhibited angiogenesis by down regulating vascular endothelial growth factor and erythropoietin expression. Moreover, vanillic acid inhibited cell proliferation via blocking cell cycle progression” to “We found that vanillic acid dose-dependently inhibited VEGF and EPO protein expressions and disrupt tube formation. The results suggest that vanillic acid effectively inhibits angiogensis. Flow cytometry analysis demonstrated that vanillic acid significantly induced G1 phase arrest and inhibited the proliferation of HCT116 cells”(Page 1, Line 23 to 27).

Reviewer 2 Report
This is generally well written and is scientifically sound with a novel line of investigation.
There are however some grammatical errors that should be corrected as outlined below:
Line 48 “the presence of 2-alphaketoglutarate. Hydroxylated HIF-1α binds to von Hippele Lindau (VHL)” – correct spelling is “von Hippel Lindau”
Line 67 “was correlated with suppression of HIF-1α protein synthesis but bot not its degradation or” – change this to “but not its degradation”
Line 342 “Based on our in vitro findings, HCT116 cells were injecting subcutaneously in mice and induced 343 tumor formation” – change this to “HCT116 cells were injected subcutaneously”
Line 355 “from five day. Tumor volume and mouse weight were measured using a equation and a disital” – correct to “an equation” and correct spelling “distal”
Line 372 “HIF-1α is a key stress-responsive transcription factor to low oxygen, the expression of HIF-1α 373 is associated with tumor progression and angiogenesis”- change this to “low oxygen; the expression…”
Line 387 “related to the activation of eIF4E. Importantly, eIF4E is a pivotal translation factor, it regulates the initiation of mRNA translation [24].” – change this to “factor; it regulates..”
Line 405 “The transcription of VEGF and EPO genes are mediated by HIF-1α, they are hypoxia-inducible 406 neuroprotective cytokines.[30].” – change this to “mediated by HIF-1a; they are..”
Line 406 “In tumor tissues, VEGF is essential for tumor angiogenesis, it can induce the new blood vessels formation [31].” – change this to “angiogenesis; it can…”
Line 410 “EPO plays an important role in tumor angiogenesis in vivo, it can promote tumor cells survival and growth under hypoxic [34].” – change this to “in vivo; it can promote..”
Line 428 “suggests the0 potential use of vanillic acid cancer treatment, and thus provides a basis for the” – correct the typo “the0”
Author Response
Thank you very much for decision letter along with the reviewer’s comments for our manuscript No.: ijms-414213
The reviewers requested modification, and more clarification. They also gave us several excellent suggestions which would strengthen our manuscript. We thank for reviewer’s constructive criticisms.
Please find some of our changes in revised MS. as highlighted in red in marked version.
Our point by point response to the comments of the reviewer is as follows:
Reviewer 2:
We have addressed the suggestions raised by the reviewer by performing a series of modifications
1. Line 48 “the presence of 2-alphaketoglutarate. Hydroxylated HIF-1α binds to von Hippele Lindau (VHL)” – correct spelling is “von Hippel Lindau”
Response 1: We thank the reviewer’s suggestions. We are very regret for our incautiousness. We have revised “von Hippele Lindau (VHL)” to “von Hippel Lindau (VHL)” (Page 2, Line 51).
2. Line 67 “was correlated with suppression of HIF-1α protein synthesis but bot not its degradation or” – change this to “but not its degradation”
Response 2: We thank the reviewer’s suggestions. We are very regret for our incautiousness. We have revised “but bot not its degradation” to “but not its degradation” (Page 2, Line 70).
3. Line 342 “Based on our in vitro findings, HCT116 cells were injecting subcutaneously in mice and induced 343 tumor formation” – change this to “HCT116 cells were injected subcutaneously”
Response 3: We thank the reviewer’s suggestions. We are very regret for our incautiousness. We have revised “HCT116 cells were injecting subcutaneously in mice” to “HCT116 cells were injected subcutaneously” (Page 13, Line 347).
4. Line 355 “from five day. Tumor volume and mouse weight were measured using a equation and a disital” – correct to “an equation” and correct spelling “distal”
Response 4: We thank the reviewer’s suggestions. We are very regret for our incautiousness. We have revised “a equation” to “an equation” (Page 14, Line 360). We have revised “disital” to “digital” (Page 14, Line 360).
5. Line 372 “HIF-1α is a key stress-responsive transcription factor to low oxygen, the expression of HIF-1α 373 is associated with tumor progression and angiogenesis”- change this to “low oxygen; the expression…”
Response 5: We thank the reviewer’s suggestions. We are very regret for our incautiousness. We have revised “low oxygen, the expression…” to “low oxygen; the expression…” (Page 15, Line 377).
6. Line 387 “related to the activation of eIF4E. Importantly, eIF4E is a pivotal translation factor, it regulates the initiation of mRNA translation [24].” – change this to “factor; it regulates..”
Response 6: We thank the reviewer’s suggestions. We are very regret for our incautiousness. We have revised “factor, it regulates…” to “factor; it regulates…” (Page 15, Line 392).
7. Line 405 “The transcription of VEGF and EPO genes are mediated by HIF-1α, they are hypoxia-inducible 406 neuroprotective cytokines [30].” – change this to “mediated by HIF-1a; they are…”
Response 7: We thank the reviewer’s suggestions. We are very regret for our incautiousness. We have revised “mediated by HIF-1α, they are…” to “mediated by HIF-1a; they are…” (Page 16, Line 411).
8. Line 406 “In tumor tissues, VEGF is essential for tumor angiogenesis, it can induce the new blood vessels formation [31].” – change this to “angiogenesis; it can…”
Response 8: We thank the reviewer’s suggestions. We are very regret for our incautiousness. We have revised “angiogenesis, it can induce…” to “angiogenesis; it can…” (Page 16, Line 412).
9. Line 410 “EPO plays an important role in tumor angiogenesis in vivo, it can promote tumor cells survival and growth under hypoxic [34].” – change this to “in vivo; it can promote…”
Response 9: We thank the reviewer’s suggestions. We are very regret for our incautiousness. We have revised “in vivo, it can promote…” to “in vivo; it can promote…” (Page 16, Line 417).
10. Line 428 “suggests the0 potential use of vanillic acid cancer treatment, and thus provides a basis for the” – correct the typo “the0”.
Response 10: We thank the reviewer’s suggestions. We are very regret for our incautiousness. We have revised “the0…” to “the…” (Page 16, Line 434).
We have re-checked the manuscript and found some inaccurate language and have made some modifications. Please review the changes in the revised MS. as highlighted in blue in marked version.
1. We are very regret for our incautiousness. We have revised “level” to “levels” (Page 1, Line 20).
2. We are very regret for our incautiousness. We have revised “declines” to “decline” (Page 2, Line 58).
3. We are very regret for our incautiousness. We have revised “isolated from the vanilla bean of Angelica sinensis as well as green tea” to “isolated from the Angelica sinensis and green tea” (Page 2, Line 64 to 65).
4. We are very regret for our incautiousness. We have revised “but does not” to “but not” (Page 9, Line 267).
5. We are very regret for our incautiousness. We have revised “HCT116 cells” to “HUVEC cells” (Page 11, Line 315).
6. We are very regret for our incautiousness. We have revised “HUVEC” to “HUVEC cells” (Page 11, Line 317).
7. We are very regret for our incautiousness. We have revised “Cell proliferation was also detected by an EdU incorporation assay. Following 12 h of treatment, vanillic acid (30 µM) decreased the number of EdU-positive cells among hypoxia-exposed HCT116 cells (Fig. 6D). EdU staining demonstrated that vanillic acid treatment significantly suppressed DNA synthesis in HCT-116 cells compared with the positive control” to “Cell proliferation was also detected by a clonogenic assay. Following 12 h of treatment, vanillic acid (30 µM) decreased the number of hypoxia-exposed HCT116 cells. In other words, the clonogenicity of HCT116 cells was reduced in a dose-dependent manner (Fig. 6D)” (Page 12, Line 329 to 332).
8. We are very regret for our incautiousness. We have revised “inhibited HIF-1α mRNA, protein synthesis” to “inhibited HIF-1α protein synthesis” (Page 15, Line 388).
9. We are very regret for our incautiousness. We have revised “Because they were induced by HIF-1α, whether vanillic acid suppressed EPO and VEGF transcription in HCT116 cells was examined further” to “Since EPO was induced by HIF-1α, we further examined whether vanillic acid suppressed EPO and VEGF transcription in HCT116 cells” (Page 16, Line 417 to 419).

Round 2
Reviewer 1 Report
Authors did not perform any further experiments based on comment 1 and 2. Lacking evidences are for conclusions.
Author Response
Dear Astrid Wang
Thank you very much for decision letter along with the reviewer’s comments for our manuscript No.: ijms-414213
The reviewers requested additional data modification, and more clarification. They also gave us several excellent suggestions which would strengthen our manuscript. We thank for reviewer’s constructive criticisms.
Please find some of our changes in revised MS. as highlighted in red in marked version.
Our point by point response to the comments of the reviewer is as follows:
Reviewer 1:
Comments and Suggestions for Authors Authors did not perform any further experiments based on comment 1 and 2. Lacking evidences are for conclusions.
Comment 1, In general, the mechanistic study of this MS is not deep despite many data. There are lacking evidences for title. How about effect of mTOR/p70S6K/4E-BP1 and Raf/MEK/ERK inhibitor or siRNA transfection on proliferation and HIF1 alpha in HCT116 cells under hypoxia?
Thanks to the reviewer’s suggestions. We have used the mTOR inhibitor (rapamycin) and MEK inhibitor (PD98059) to examine HIF-1α expression level and clonogenicity in HCT116 cells. The results showed that the vanillic acid (30 μM), or rapamycin (100 nM) suppressed the expression of HIF-1α, which was further inhibited by the combination of vanillic acid and rapamycin (Fig. 4C and 4G). Comparable results were obtained in Fig. 4D and 4H, vanillic acid (30 μM) or PD98059 (50 μM) suppressed the expression of HIF-1α, which was further inhibited by the combination of vanillic acid and PD98059.
Fig. 6F results showed that vanillic acid (30 μM) or rapamycin (100 nM) suppressed clonogenicity of HCT116 cells, which was further inhibited by the combination of vanillic acid and rapamycin. Comparable results were obtained in Fig. 6G, vanillic acid (30 μM) or PD98059 (50 μM) suppressed clonogenicity of HCT116 cells, which was further inhibited by the combination of vanillic acid and PD98059. And the results are as follows.
Comment 2, Show antiproliferative effect of vanillic acid in other colon cancer lines under hypoxia. You just showed its effect only on HCT116 cells?
We thank the reviewer’s suggestions. We have examined the effect of vanillic acid on proliferative in other colon cancer lines by performed western blot assay and clonogenic assay. The results showed that vanillic acid dose-dependently reduced HIF-1α protein expression induced by 1% O2 in SW620 cells (Fig. 2C and Fig. 2F). Fig. 6E results showed that vanillic acid significantly inhibted the clonogenicity of SW620 cells. And the results are as follows.

Round 3
Reviewer 1 Report
Much improved